# Long-term continuous no-till corn-soybean systems: Examining soil carbon sequestration and nitrogen accumulation across various pools

**Maninder K. Khosa**[1☯], **Kenan Barik**[2☯], **Ekrem Aksakal**[2☯], **Mohammad MR. Jahangir**[3☯], **Nataliia O. Didenko**[4☯], **Khandakar R. Islam**[5☯]*

**1** Dept. of Soils, Punjab Agricultural University, Ludhiana, India, **2** Dept. of Soil Science and Plant Nutrition, Ataturk University, Erzurum, Turkey, **3** Dept. Soil Science, Bangladesh Agricultural University, Mymensingh, Bangladesh, **4** Institute of Water Problems and Land Reclamation, Kyiv, Ukraine, **5** Soil, Water and Bioenergy Resources, Ohio State University South Centers, Piketon, United States of America

☯ All the authors contributed equally and rationally to this work.

\* islam.27@osu.edu

## Abstract

Tillage practices influence the soil's capacity as either a sink or source of carbon (C) within agroecosystems. The objective of the study was to assess the effects of no-till (NT) management over 0, 6, 20, and 35 years in a rainfed corn (*Zea mays*)–soybean (*Glycine max*) system, incorporating cereal rye (*Secale cereale* L.) as a cover crop, on soil organic C (SOC) sequestration and total N accumulation across different pools. The results showed significant increases under NT compared to conventional tillage (CT), including SOC (14–69%), total N (16–60%), microbial biomass C (SMB; 44–101%), active C (11–33%), passive C (15–72%), particulate organic C (POC; 43–173%), and particulate organic N (PON; 29–135%). While NT exhibited higher basal respiration (BR) rates, it significantly reduced C loss via the metabolic quotient, calculated as specific maintenance respiration ($qCO_2$), by 7.9–29.4% compared to CT. Ratios of passive C to active C increased under long-term NT, indicating a higher accumulation of stable SOC fraction, which consequently reduced soil bulk density ($\rho b$) compared to CT. Using the fixed depth approach, SOC, SMB, active C, and POC were sequestered at rates of $587.4 \pm 134.1$, $10.1 \pm 4.1$, $5.3 \pm 1.8$, and $382.2 \pm 55$ kg ha$^{-1}$ yr$^{-1}$ in the 0–15 cm depth, and at $1.6 \pm 0.5$, $4.1 \pm 1.4$, $54 \pm 8$, and $192 \pm 64$ kg ha$^{-1}$ yr$^{-1}$ in the 15–30 cm depth. Likewise, total N and PON accumulation rates were $72.2 \pm 18.4$ and $14.1 \pm 5.5$ kg ha$^{-1}$ yr$^{-1}$ at 0–15 cm, and $15 \pm 5$ and $4.3 \pm 1.6$ kg ha$^{-1}$ yr$^{-1}$ at 15–30 cm. Similar but variable rates of SOC sequestration and total N accumulation were observed at both depths when using the equivalent mass approach compared to the fixed depth method. Adjusting for soil mass equivalence to account for $\rho b$ variability in fixed depths provides a more realistic estimation of SOC and total N stocks in different pools, as the fixed depth approach tends to overestimate these stocks. Our findings demonstrate that long-term NT consistently facilitates SOC

**Data availability statement:** All relevant data are available within the paper and its Supporting Information files.

**Funding:** The author(s) received no specific funding for this work.

**Competing interests:** The authors have declared that no competing interests exist.

sequestration and total N accumulation in different pools, with these benefits distributed non-linearly across distinct SOC and total N pools at the 0–15 cm depth and linearly at the 15–30 cm depth in rainfed corn-soybean systems.

## Introduction

Conservation agriculture has the potential to rejuvenate and maintain soil quality by sequestering atmospheric $CO_2$ through plants into soil organic matter (SOM). Soil, which comprises 61% of the largest C pool in terrestrial ecosystems, faces continuous depletion due to human activities [1–2]. Approximately 1.5 Pg $CO_2$ is emitted annually from deforestation, grassland degradation, and conventional agricultural practices, with annual tillage being a major contributor to SOC and N loss worldwide [3–4].

Conventional tillage (CT), a traditional method for land preparation and crop production, offers benefits such as soil inversion, mixing of crop residues, reduced surface compaction, increased air circulation, weed and disease control, and improved field workability [5–6]. However, greater reliance on CT degrades soil quality by affecting biodiversity, disrupting structural stability, increasing decomposition of crop residue and native SOM, and accelerating erosion [6–7]. Accelerated macroaggregate dispersion caused by CT exposes particulate organic matter (POM), leading to its mineralization and subsequent release of $CO_2$ into the atmosphere, along with available nitrogen (N) in the soil [8–9]. A meta-analysis indicated that CT provides a higher amount of available N through mineralization compared to no-till (NT), particularly in the absence of N fertilization [10]. Furthermore, the inversion and mixing of soil during CT create more aerobic and warmer conditions, which accelerate the decomposition of fragmented crop residues and the mineralization of native soil organic carbon (SOC) by opportunistic microbes. These processes significantly influence the biological, chemical, and physical properties associated with soil quality [11–13].

Transitioning from CT to NT, a key climate-smart agricultural practice, is effective in reducing operating costs, minimizing soil erosion, conserving moisture, and maintaining the soil's physical stability [14–15]. Long-term NT is reported to enhance biodiversity and nutrient recycling, improve soil quality by accumulating SOM, and mitigate climate change by reducing greenhouse gas emissions from agriculture [12,14,16]. In 2021, NT was practiced on over 44.5 million ha in the United States, up from 42 million ha in 2017 and 1.34 million ha in the early 1970s [17]. Globally, NT covered about 180 million ha, representing approximately 12.5% of total global cropland [15,17]. The adoption of NT is increasing worldwide, driven by farm economics, time savings, enhanced agroecosystem services, and potential C credits through SOM accumulation [17].

Soil organic matter is a composite indicator of soil quality, serving as a reservoir of labile C substrates, energy, enzymes, and nutrients, particularly N [8,18,19]. Both SOC and total N are critical for ecosystem functioning and productivity, as they maintain biodiversity and efficiency, regulate biochemical processes, support

physical stability and hydrological functions, and enhance agricultural resilience [5,19]. The SOC exists in diverse fractions, and treating it as a simple homogeneous pool overlooks its variations in chemical composition, lability, biochemical turnover, and physicochemical stability [18]. Significant changes in SOC content typically require years to detect due to its inherent biochemical complexity and interactions with clays and calcium. However, measuring labile C fractions can provide earlier and more consistent indications of management-induced changes in SOC [18,20]. Labile C, defined as a small SOC fraction with rapid turnover rates, is more readily utilized by microbes compared to bulk SOC [5,18,21]. Labile C fractions include microbial biomass C (SMB), active C, particulate organic C (POC), potentially mineralizable C (PMC), light-fraction C, cold and hot water-extractable C, anthrone-reactive C, and fulvic acid C [2,20]. Various metabolic quotients and indices, such as basal respiration (BR), SMB: SOC (qR), microbial quotient (specific maintenance respiration, $qCO_2$), and C management index, have been identified as sensitive indicators of labile C fractions [5]. These fractions are the primary sources of SOC losses associated with CT [5,7,20].

Several studies have reported that changes in labile C are often associated with corresponding changes in labile N due to the stoichiometric relationship between C and N in SOM [22]. These changes can influence soil biotic activity, the decomposition of residues and native SOM, and, consequently, the sequestration of SOC and the accumulation of total N [20,23]. As the majority of soil N is associated with SOM in constant proportions, an increase in SOM typically corresponds to an increase in soil N [24]. Although total N is biochemically more labile than SOC, it is expected that total N will exhibit more pronounced changes compared to SOC in response to change in management practices [22,25]. Sensitive indicators of labile N fractions, such as active N, soil microbial biomass N, potentially mineralizable N (PMN), available N, particulate organic N (PON), and the nitrogen management index (NMI), have been widely studied [5,26,27].

Parent material and soil textural diversity inherently influence SOC sequestration and total N accumulation [28]. However, the conversion of CT to NT is expected to impact SOC sequestration and total N accumulation across both labile and non-labile fractions. Several studies have demonstrated that NT adoption under humid or subhumid climates, particularly in medium- and heavy-textured soils, promotes greater SOC sequestration compared to CT [29–30]. Conversely, in warmer, drier climates and sandy soils, NT has been reported to result in negligible SOC gains compared to CT [30–31]. While many studies have shown that NT significantly increases SOC content in surface soil, CT maintains SOC levels at deeper soil layers below 20 cm [12,32,33]. Nevertheless, some research indicates that long-term NT may enhance SOC concentrations at deeper depths due to processes such as the downward movement of crop residues facilitated by earthworms and soil fauna, leaching of dissolved organic carbon, and extended root growth [29,34]. Limited oxygen diffusion in moist, cooler, and partially anaerobic soils under NT may also restrict root decomposition, leading to greater root-derived SOC accumulation in deeper soil layers.

Studies have reported that NT significantly increases SOC along with SMB, active C, and POC contents, particularly within the 0–15 cm soil depth, compared to CT. Similarly, N dynamics are influenced by tillage practices, with NT resulting in 24–34% higher total N levels than CT due to the accumulation of SOM [35]. After 48 years of NT, total N stocks in the top 20 cm increased by approximately 1000 kg N ha⁻¹ compared to CT systems [35]. A meta-analysis further demonstrated that long-term NT enhanced PMN compared to plowing, attributed to greater total N stocks [36]. Recent findings [37] suggested a 50% increase in PON stock within the 0–30 cm depth under NT (4.43 Mg N ha⁻¹) compared to CT (2.96 Mg N ha⁻¹). As indicators of the labile fraction, these SOC and total N fractions provide valuable insights into the long-term impacts of tillage practices on the stability and stocks of SOC and total N.

Despite NT's importance in agroecosystem services, its impact on SOC sequestration and total N accumulation in different fractions is still debated when accounting for soil depth and bulk density ($\rho b$) variations [3,4,38]. Temporal variations in tillage operations that affect SOM via C and N contents also impact the depth distribution of $\rho b$. While SOC sequestration and total N accumulation is often quantified by comparing SOC and total N stocks within a fixed soil depth (commonly 30 cm), this method can be inadequate and prone to bias due to variations in $\rho b$, which may lead to the overestimation or underestimation of SOC and total N stocks [38–39]. Differences in SOC and total N stocks associated with changes in $\rho b$

can be corrected using the equivalent soil mass (ESM) approach, which accounts solely for variations in SOC and total N concentrations [40–42]. The ESM approach is therefore crucial for accurately calculating or predicting SOC sequestration and total N accumulation associated with SOM dynamics, particularly when comparing management practices [43–44].

Although the significance of ESM for assessing SOC stock dynamics has been emphasized, it is not yet widely implemented, particularly in evaluating C credits derived from agricultural SOC sequestration [38,39,44]. Furthermore, understanding changes in SOC sequestration and total N accumulation under different tillage practices is essential for determining optimal N fertilization rates [45]. Despite substantial research on SOC and total N in managed agroecosystems, a knowledge gap remains concerning the effects of long-term NT practices on SOC and total N contents across different fractions and soil depths [7].

We hypothesized that long-term continuous NT practices would result in significant depth-specific differences in SOC, total N, and their labile fractions compared to CT in farmers' fields under temperate climates with similar crop rotation and climatic conditions. The objectives of our on-farm study were to evaluate the effects of continuous NT in rainfed corn-soybean systems with cereal rye as a cover crop (0, 6, 20, and 35 years) on (1) SOC, total N, SMB and associated biological properties, including BR, $qCO_2$, and PMC, as well as active C, passive C, POC, and PON concentrations, (2) soil bulk density ($\rho b$), and (3) SOC sequestration and total N accumulation in different pools at 0–15 cm and 15–30 cm soil depths, comparing fixed-depth and ESM approaches to CT as a control.

## Materials and methods

### Site description

The study was conducted at the 495-ha Brandt's Family farm in Carrol (39°50′915″ N, 82°41′579″ W), Fairfield County, in central Ohio, United States (Fig 1). It is one of the oldest continuous NT and cover crops adapted farms in Ohio since 1971 [7]. The topography of the farm is moderate, with surface inclinations ranging between 1 and 3 degrees. Throughout the years, the farm site experiences a humid continental climate with four distinct seasons. Winters are cold and snowy, with temperatures averaging between -3.9°C and 0°C with snowfall averages 48.3 cm annually. Summers are hot and humid, with average temperatures ranging from 18.3°C to 26.7°C, with highest in July. Spring and fall offer mild temperatures. Rainfall is fairly consistent year-round, averaging around 102 cm, with spring being the wettest season.

Bennington silt loam (fine, illitic, mesic Aeric Epiaqualf) is the dominant soil across all sampled NT fields [46]. Averaged across NT fields, soil baseline properties at a depth of 0–15 cm include a pH of 5.6 to 6.2, electrical conductivity (ECe) of 158–283 dS/m, cation exchange capacity (CEC) of 15–24 meq/100 g, 2M KCl extractable N of 17.8 to 27.1 mg/kg, Mehlich-III extractable phosphorus of 10–24 mg/kg, sand 23–26%, silt 58–61%, and clay 15–17%. At a depth of 15–30 cm, soil properties include a pH of 5.5 to 5.9, ECe of 17–21 dS/m, CEC of 16–21 meq/100 g, extractable N 6–14 mg/kg, extractable P of 5–9 mg/kg; sand 20–22%, silt 59–60%, and clay 18–21%.

### Soil sampling and processing

Soil samples were collected from fields with six ($NT_6$), twenty ($NT_{20}$), and thirty-five ($NT_{35}$) years of continuous NT management, as well as from an adjacent CT field ($NT_0$) on Brandt's farm. Over the years, the crop rotation was a rainfed corn-soybean system with cereal rye as a winter cover crop used at all fields. The NT fields utilized a planter with a wavy coulter to open a 5 cm wide path for planting crops without disturbing the surface residues. In contrast, the CT plots were chisel plowed annually in the fall. All plots received standard chemical fertilization and weed control practices. Nitrogen fertilization was performed using 28% urea-ammonium nitrate (UAN) at 160 L ha$^{-1}$ at planting, followed by 28% UAN at 455 L ha$^{-1}$ as a side-dress. Phosphorus and potassium were broadcast annually at 100 kg ha$^{-1}$ using 0-46-0 and 0-0-61 chemical fertilizers in the previous fall.

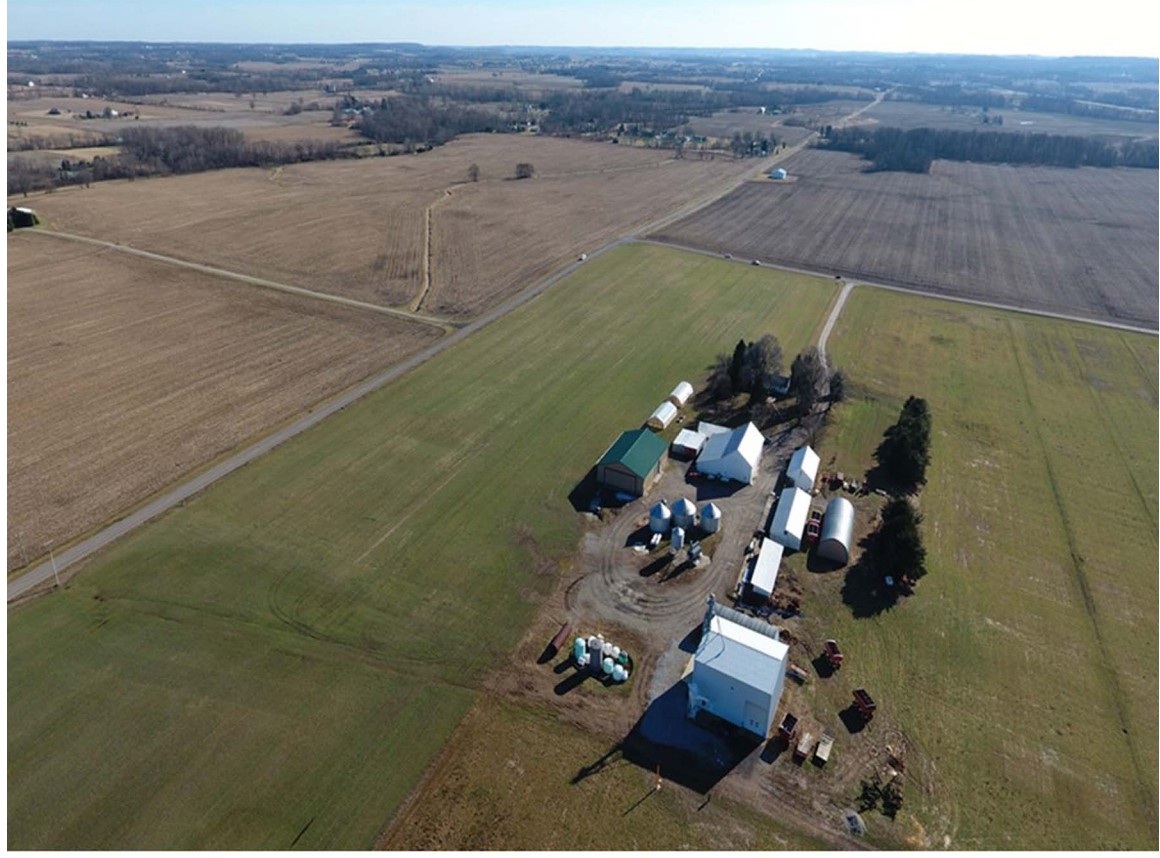

**Fig 1. Aerial view of the Brandt's family farm at Carroll, Fairfield county, Oho, USA.**

The global positioning system (GPS) referenced four plots (20 m x 20 m) as pseudo-replicates within each NT and CT field were established before soil sampling. Six cores were collected from a depth of 0–30 cm (segmented into 0–15 and 15–30 cm) using a Giddings soil probe from each of the GPS-referenced pseudo-replicated plots, following a systematic sampling procedure (4 NT treatments x 4 pseudo-replicated plots x 2 depths x 6 cores = 192 cores). The cores were pooled and mixed to obtain a composite sample at each depth and placed in sealable plastic bags (4 NT treatments x 4 pseudo-replicated plots x 2 depths = 32 samples). A sub-sample of the field-moist soil was sieved to 2 mm for analysis of SMB, associated biological properties (BR and PMC), and extractable N and P. Another portion of the soil was air-dried in the shade at room temperature (~25°C) for 15 days before being analyzed for SOC, total N, active C, POC, PON, pH, ECe, CEC, and particle size distribution.

### Soil analysis

The SMB was determined using a rapid microwave irradiation and extraction method [47]. Basal respiration (BR), as a measure of antecedent biological activity, was determined using *in vitro* static incubation of unamended field-moist soil at $25 \pm 1°C$ in a temperature-controlled incubator for a period of 15 days [8]. Specific maintenance respiration ($qCO_2$), the ecophysiological or microbial metabolic quotient, was calculated as BR per unit of SMB per day [8,48]. The PMC was calculated by dividing the total amount of $CO_2$ evolved from the soil during the 15 days *in vitro* incubation by the SOC content [5].

The SOC and total N contents were measured on finely ground (<200 μm) air-dried soils using the dry combustion method with a FlashEA-1112 series CNHS-O analyzer°. Active C was determined spectrophotometrically based on mild oxidation of SOC in 2-mm sieved air-dried soil with a pH neutral 0.02M $KMnO_4$ solution [18]. Passive C was calculated by subtracting active C from SOC. The SMB: SOC ratio, also known as the microbial quotient (qR), was calculated [48].

To determine POC and PON contents, free- and aggregate-protected POM was collected after dispersing a 10 g sample of 2-mm air-dried soil with small glass marble in 50-mL screw-top polypropylene tubes by shaking with 25 mL of 0.5% Na-hexametaphosphate solution at 250 rpm for 18 hr. [49]. After shaking, the dispersed soil suspensions were passed through a 53-μm sieve, washed slowly under running water to collect sand-associated POM, and oven-dried at 65±2°C using a forced-air oven until a constant weight was obtained. A sample of sand-associated POM was burned at 480±10°C for 4 hr. in a muffle furnace to calculate POM content by loss on ignition. Another sample of finely ground sand-associated POM was placed in Zn capsules and analyzed for POC and PON concentrations using the FlashEA-1112 series CNHS-O analyzer°. Results were expressed on a sand-free basis.

Soil pH was determined in 1:2.5 soil-distilled deionized water slurries using a combination glass electrode. The ECe was determined in 1:1 soil-distilled deionized water using a standard conductivity meter. Antecedent soil moisture content was determined thermogravimetrically after oven-drying the field-moist soil at 105±2°C for 24 hr. until a constant weight was obtained. The AN with 2M KCl extraction and Mehlich-3 AP were determined by Shimadzu° Dissolved TOC/TOC analyzer and Integrated Coupled Plasma Emission (ICPE) Spectrophotometry, respectively. Soil CEC was determined following the ammonium acetate extraction and Shimadzu° Dissolved TOC/TOC analyzer.

Soil ρb was calculated by the standard core method from the relationship of oven-dried mass of a known volume of soil. Particle size analysis for sand, silt, and clay contents was determined by the Bouycous hydrometer method.

## Soil carbon sequestration and nitrogen accumulation

Using the fixed-depth approach, the stocks of SOC, total N, SMB, active C, passive C, POC, and PON were calculated. These calculations involved multiplying their respective concentrations by the sampling depth (d) and the corresponding ρb values, as concurrently measured [41–42].

$$\text{Fixed depth stock}(\text{Mg ha}^{-1}) = (\rho b \times d \times C \text{ or N pools} \times 10000)$$

Soils are sensitive to changes in ρb due to transitional or long-term tillage practices, where an increase or decrease in ρb invariably leads to a corresponding change in soil mass for a given depth over time. To account for this, it is essential to adjust for soil depth to minimize random errors associated with variations in ρb. A depth correction factor ($d_{eq}$) was calculated by dividing the lowest ρb value by the ρb values of all treatments. This correction factor was then applied before converting SOC and total N stocks to an equivalent mass basis [41–42]:

$$\text{Equivalent mass stock } (\text{Mg ha}^{-1}) = (\rho b \times C \text{ or N pools} \times 1000) \times d_{eq} \text{ factor}$$

To evaluate the capacity of NT soils to sequester SOC and accumulate total N in various fractions at 0–15 cm and 15–30 cm depths, fixed-depth and equivalent mass stocks of SOC, total N, SMB, active C, passive C, POC, and PON were plotted as a function of NT duration (0, 6, 20, and 35 years) using best-fit linear and non-linear regression models.

Additionally, the equivalent mass stocks of SOC, total N, SMB, active C, passive C, POC, and PON were regressed against their fixed-depth counterparts to address variability and to account for potential overestimation or underestimation in predicting the capacity of NT soils to sequester SOC and accumulate total N in both bulk and labile pools.

## Statistical analysis

Significant differences in SOC and total N concentrations, stocks, and sequestration and/or accumulation across different fractions were attributed to the combined effects of years under no-till (NT) management, soil depth, and their interactions. These differences were evaluated using a factorial experimental design comprising four NT treatments and two soil depths, analyzed with the general linear modeling (GLM) procedure for two-way analysis of variance in SAS®. To ensure the validity of the GLM procedure, the normality assumption for residuals was assessed through a normal probability plot and verified using the Shapiro–Wilk test. All dependent variables satisfied the assumptions of normality and homogeneity of variances.

While soil sampling and analyses were conducted as a single event in fields managed under NT for varying durations, rather than as repeated measures over time in the same fields, NT and soil depth were considered fixed predictor variables. The main and interaction effects of NT treatments and soil depth on dependent variables were analyzed using the Least Significant Difference (LSD) test, with statistical significance determined at $p \leq 0.05$, unless stated otherwise. To further quantify SOC sequestration and total N accumulation within different soil pools, best-fit linear and non-linear regression models were applied. These estimations were carried out separately for the 0–15 cm and 15–30 cm soil depth intervals, with calculations performed using SigmaPlot® software.

## Results and discussion

### Changes in total and labile soil organic carbon and nitrogen contents

Tillage and soil depth influenced SOC, total N, and their labile fractions, with no significant interactions observed. (Tables 1 and 2). No-till treatments had significantly higher SOC (by 14–69%) and total N (16–60%) contents than CT,

**Table 1. Long-term continuous no-till effects (0, 6, 20, and 35 yr.) on total soil organic carbon (SOC), total nitrogen (total N), microbial biomass carbon (SMB), microbial quotient (qR), basal (BR) respiration and specific maintenance ($qCO_2$) respiration rates, and potentially mineralizable carbon (PMC) at different soil depths under a rainfed corn-soybean system with cereal rye as a winter cover crop.**

| Tillage Trt. | Depth (cm) | SOC (g/kg) | Total N | SMB (mg/kg) | qR (%) | BR (mg/kg/d) | $qCO_2$ (µg/mg/d) | PMC (%) |
|---|---|---|---|---|---|---|---|---|
| $NT_0$ (CT) | | 12.1c[≠] | 1.12c | 123.2c | 1.06c | 14.7b | 117.8a | 1.83b |
| $NT_6$ | | 13.8c | 1.3bc | 176.1b | 1.32a | 19a | 108.5b | 2.16a |
| $NT_{20}$ | | 16.5b | 1.53b | 189.3b | 1.16b | 19.3a | 99.4c | 1.7bc |
| NT35 | | 20.5a | 1.79a | 248.2a | 1.2b | 21a | 83.2d | 1.5c |
| Soil depth | | | | | | | | |
| | 0-15 | 20.8x[δ] | 1.85x | 235.7xd | 1.11x | 24.7x | 110.2x | 1.80x |
| | 15-30 | 10.7y | 1.02y | 132.7y | 1.26y | 12.3y | 94.2y | 1.80x |
| Tillage x soil depth | | | | | | | | |
| $NT_0$ (CT) | 0-15 | 15.5ns | 1.34ns | 139.3ns | 0.9ns | 18.7ns | 135.9ns | 1.81ns |
| | 15-30 | 8.7 | 0.9 | 107.1 | 1.23 | 10.7 | 99.7 | 1.85 |
| $NT_6$ | 0-15 | 18.6 | 1.76 | 222.9 | 1.2 | 23.3 | 106.1 | 1.89 |
| | 15-30 | 9 | 0.83 | 129.3 | 1.42 | 14.7 | 110.8 | 2.43 |
| $NT_{20}$ | 0-15 | 21.8 | 2 | 246.2 | 1.13 | 27.3 | 111.1 | 1.88 |
| | 15-30 | 11.2 | 1.07 | 132.4 | 1.18 | 11.3 | 87.6 | 1.51 |
| $NT_{35}$ | 0-15 | 27.3 | 2.31 | 334.4 | 1.22 | 29.3 | 87.6 | 1.61 |
| | 15-30 | 13.7 | 1.27 | 162 | 1.18 | 12.7 | 78.8 | 1.39 |

[δ]Treatment means under each column separated by same lower-case letters (x and y) were not significantly between soil depths at $p \leq 0.05$.

[≠]Treatment means under each column separated by same lower-case letters (a, b, c, and d) were not significantly different among years of NT at $p \leq 0.05$.

**Table 2. Long-term continuous no-till (NT) effects (0, 6, 20, and 35 yr.) on active carbon (AC), passive C (PC), particulate organic carbon (POC), and particulate organic nitrogen (PON) contents at different soil depths under a rainfed corn-soybean system with cereal rye as a winter cover crop.**

| Tillage Trt. | Depth (cm) | AC (mg/kg) | PC (g/kg) | PC: AC | POC (g/kg) | PON | POC: PON | POC: SOC | PON: TN |
|---|---|---|---|---|---|---|---|---|---|
| $NT_0$ (CT) | | 454.3c[≠] | 11.6c | 25.6b | 2.66d | 0.122d | 22.1a | 22c | 10.9b |
| $NT_6$ | | 504.2b | 13.3c | 26.4b | 3.84c | 0.158c | 24a | 27.8b | 12.2b |
| $NT_{20}$ | | 588.1a | 15.9b | 27.1b | 5.57b | 0.238b | 23.5a | 33.8a | 15.6a |
| $NT_{35}$ | | 603.9a | 19.9a | 30.8a | 7.27a | 0.287a | 22.5a | 35.5a | 16.0a |
| Soil depth | | | | | | | | | |
| | 0-15 | 606.6x[δ] | 20.2x | 33.0x | 7.58x | 0.487x | 15.5x | 36.4x | 26.3x |
| | 15-30 | 468.7y | 10.2y | 21.6y | 2.09y | 0.136y | 15.5x | 19.5y | 13.3y |
| Tillage x soil depth | | | | | | | | | |
| $NT_0$ (CT) | 0-15 | 504.9ns | 15ns | 29.7ns | 3.9ns | 0.183ns | 21.3ns | 25.2ns | 13.7ns |
| | 15-30 | 403.7 | 8.3 | 20.6 | 1.42 | 0.062 | 22.9 | 16.3 | 6.9 |
| $NT_6$ | 0-15 | 601.6 | 18 | 29.9 | 5.93 | 0.241 | 24.6 | 31.9 | 13.7 |
| | 15-30 | 406.9 | 8.4 | 21.1 | 1.75 | 0.075 | 23.4 | 19.4 | 9.0 |
| $NT_{20}$ | 0-15 | 648.1 | 21.1 | 32.6 | 8.69 | 0.373 | 23.3 | 39.9 | 18.7 |
| | 15-30 | 528.2 | 10.7 | 20.2 | 2.44 | 0.103 | 23.6 | 21.8 | 9.6 |
| $NT_{35}$ | 0-15 | 671.8 | 26.6 | 35.7 | 10.2 | 0.452 | 22.3 | 37.4 | 19.6 |
| | 15-30 | 535.9 | 13.2 | 24.6 | 2.76 | 0.122 | 22.6 | 20.1 | 9.6 |

[≠]Treatment means under each column separated by same lower-case letters (a, b, c, and d) were not significantly different among years of NT duration at $p < 0.05$

[δ]Treatment means under each column separated by same lower-case letters (x and y) were not significantly between soil depths at $p \leq 0.05$.

with the highest under long-term $NT_{35}$. Higher SOC and N contents under NT stoichiometrically increased labile fractions of C and N contents and associated biological properties, most visible in SMB, BR, $qCO_2$, AC, POC, and PON values, when compared to CT.

The SMB, a biologically labile fraction of SOC, was significantly highest under $NT_6$, $NT_{20}$, and $NT_{35}$, with increases of 42.9%, 53.6%, and 101%, respectively, compared to CT (Table 1). However, the qR, a microbial quotient representing the size of the biological pool of SOC, increased inconsistently, with the highest values under $NT_6$ (1.32%), followed by $NT_{20}$ (1.16%) and $NT_{35}$ (1.2%), and the lowest under CT (1.06%). Like SMB, the BR rates increased by 29, 31, and 43% under $NT_6$, $NT_{20}$, and $NT_{35}$, respectively, compared to CT. In contrast, $qCO_2$, a measure of microbial catabolism, was higher under CT and reduced by 7.9, 15.6, and 29.4% under $NT_6$, $NT_{20}$, and $NT_{35}$, respectively. The PMC showed the highest values under $NT_6$ and the lowest under $NT_{35}$ compared to CT.

Both active and passive C contents under years of NT were higher than those under CT, with more passive C accumulating under long-term NT (Table 2). The highest active C content was observed under $NT_{35}$ (increased by 33%), and the lowest under $NT_6$ (increased by 11%), compared to CT. Similarly, $NT_{35}$ had the highest passive C content (increased by 72%), followed by $NT_{20}$ (increased by 37%) and $NT_6$ (increased by 15%) compared to CT. The $NT_{35}$ exhibited the significantly highest passive C: active C ratio, followed by $NT_{20}$, $NT_6$, and CT. Both POC and PON progressively increased under years of NT compared to CT, with the lowest increase in POC and PON observed under short-term $NT_6$ (44% and 29%, respectively), and the highest under long-term $NT_{35}$ (173% and 135%, respectively). While SOC as POC significantly increased by 27.8 to 35.5% under NT compared to 22% under CT, total N as PON increased by 12.2 to 16% under NT compared to 10.9% under CT.

The SOC, total N, and their labile fractions (SMB, AC, POC, and PON), along with associated biological properties (BR and $qCO_2$), exhibited soil depth stratification when averaged across NT treatments (Tables 1 and 2). While soil depth

stratification of SOC, total N, SMB, BR, $qCO_2$, AC, POC, and PON significantly increased by 94, 81, 78, 101, 17, 29, 263, and 258%, the qR decreased by 14%, and PMC did not show such stratification. Long-term NT enhanced the stratification of SOC, total N, SMB, BR, $qCO_2$, AC, POC, and PON, while reducing the stratification of qR.

By accounting for similar soils, crop rotation, and climatic conditions, the significantly higher SOC and total N contents under NT compared to CT were attributed to the surface accumulation of unfragmented crop residues under moist, partially anaerobic, cooler, and undisturbed conditions, as well as greater root growth and chemical fertilization [8,12]. Due to reduced contact between microbes and surface-placed crop residues, the residues decompose slowly through a fungi-dominated food web with higher C-use efficiency, promoting longer retention of crop residues and enhancing bio-physical SOC sequestration, chemical stabilization, and partitioning among different pools [20,29]. In contrast, the reduced SOC and total N contents under CT were associated with greater physical disturbance from tillage and intimate mixing of fragmented crop residues in warmer, aerobic soil conditions, leading to accelerated decomposition by energy efficient bacteria and opportunistic microbes with lower C-use efficiency [29,50]. Frequent tillage can break down soil macroaggregates and expose SOM to accelerated microbial oxidation [51]. The increase in SOC content under NT or decrease under CT influenced total N content, or vice versa, due to the C and N stoichiometric coupling in SOM [25].

Significantly higher SMB, qR, and BR, along with improved biological efficiency ($qCO_2$) under NT compared to CT, were associated with greater accumulation of SOC as substrates and total N as nutrients [50,52,53], higher anabolism, and the absence of soil physical disturbance [50,54]. The higher anabolism (C-use efficiency) under long-term NT supported a larger SMB pool with lower $qCO_2$ rates compared to CT [8,33,52]. Lower $qCO_2$ values indicate that SMB are less stressed and assimilate labile C more efficiently for cell proliferation, explaining the larger SMB pool under NT. In contrast, greater physical disturbance and reduction in labile C are expected to negatively impact SMB growth and development, thus reducing metabolic efficiency [20,55]. Microbial C-use efficiency is typically higher in less stressed ecosystems [56], where SMB are more abundant and metabolically active, as evidenced by the correlation among BR, $qCO_2$, and SMB (Fig 2).

Higher contents of both POC and PON under long-term NT compared to CT, derived from partially fragmented to unfragmented crop residues, particularly roots, as well as microbially-processed C and N metabolites, contributed to macroaggregate formation and were protected within aggregates from opportunistic microbial decomposition [12,29,54]. Long-term NT resulted in greater surface stratification of SOC and total N including their labile fractions, providing food and energy sources that enhanced SMB growth and activity to enmesh microaggregates and primary soil particles bio-physically into macroaggregates compared to deeper soils. Additionally, a significantly higher content of both POC and PON at NT surface soil is due to increased root biomass distribution to form macroaggregates and get protected within aggregates [52,57,58].

Long-term retention of unfragmented residues under NT surface, compared to intimately mixed fragmented residues under CT, exposes residues to direct sunlight and UV radiation, which alters residue quality and slows decomposition [20,59]. This results in a higher proportion of passive C to active C accumulation in SOC [20]. Consequently, the passive C to active C ratio in SOC is higher, supporting our findings that SOC accumulates due to greater biochemical and physical stability under long-term NT. Passive C is a biochemically processed and physically protected pool of SOC, with lower susceptibility and accessibility to SMB decomposition [56,60,61]. Moreover, the greater stabilization of SOC under NT is linked to microbially-processed C metabolites derived from roots [61].

Significant depth stratification of SOC, total N, SMB, BR, $qCO_2$, active C, passive C, POC, and PON, except for qR values, occurred due to the greater availability of labile C and N substrates that support more efficient microbial activities in surface soil (0–15 cm depth) compared to sub-surface soil (15–30 cm depth).

## Changes in soil bulk density

Soil bulk density (ρb) was variably influenced by the duration of NT (Fig 3a). Initially, ρb increased by 3.4% under short-term NT ($NT_6$), but then decreased by 4.7% and 7.1% under long-term $NT_{20}$ and $NT_{35}$, respectively. This decrease in ρb

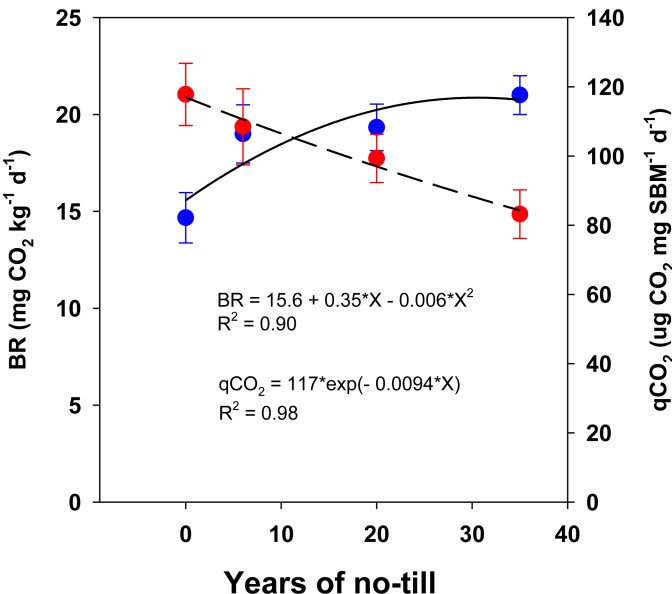

**Fig 2. Relationship of soil microbial biomass carbon (SMB) with basal respiration (BR) and specific maintenance respiration (qCO$_2$) rates across soil depths under no-till rainfed corn-soybean system with cereal rye as a winter cover crop.** Treatment mean values are presented along with the standard error of the mean.

was non-linear and more pronounced in the surface soil (0–15 cm depth) compared to the sub-surface soil (15–30 cm depth), where the decrease was linear.

The relationship of ρb with years of NT have accounted for 53% and 88% of the variability ($R^2$) in ρb changes at 0–15 and 15–30 cm depths, respectively. In addition, when ρb values were regressed on SOC contents, the significant non-linear relationship accounted 82% of the variations in decreasing ρb by increasing SOC content under NT (Fig 3b).

Increased ρb under short-term NT is due to temporary consolidation and compaction of soil by natural processes [29]. However, long-term NT reverses this effect by increased SOC content followed by macropore formation by soil fauna especially earthworms and plant roots [62]. Previous studies report a 10% increase in ρb under transitional to short-term NT, but long-term NT shows comparable or lower ρb than CT due to increased SOC content [29]. SOM, as the prime source of SOC, with its low ρb values (0.5 to 0.6 g/cm³) and ability to enhance aggregate formation, increases macroaggregation and decreases ρb under long-term NT [63]. A higher subsurface soil ρb compared to surface soil was anticipated due to lower SOC contents and macroaggregate formation. This expectation is supported by a significant non-linear decrease in ρb corresponding with an increase in SOC content under long-term NT management, corroborating findings from previous NT studies [29].

## Changes in soil organic carbon sequestration and nitrogen accumulation

The stocks of SOC and total N in different pools, calculated by multiplying their respective concentrations with depth and antecedent ρb using soil fixed depth and equivalent mass approaches, were significantly influenced by NT duration (Figs 4-7). The stocks of SOC and total N, along with their labile fractions, calculated based on the fixed depth, exhibited surface (0–15 cm depth) stratification. However, under long-term NT, the differences in their stocks between depths gradually diminished, showing a significant exponential increase toward a plateau over the years. This trend was observed for most SOC and total N fractions, except for active C, which exhibited a linear increase. When using the equivalent mass, a moderately variable impact of NT on SOC and N stocks was observed at both depths compared to the fixed depth basis.

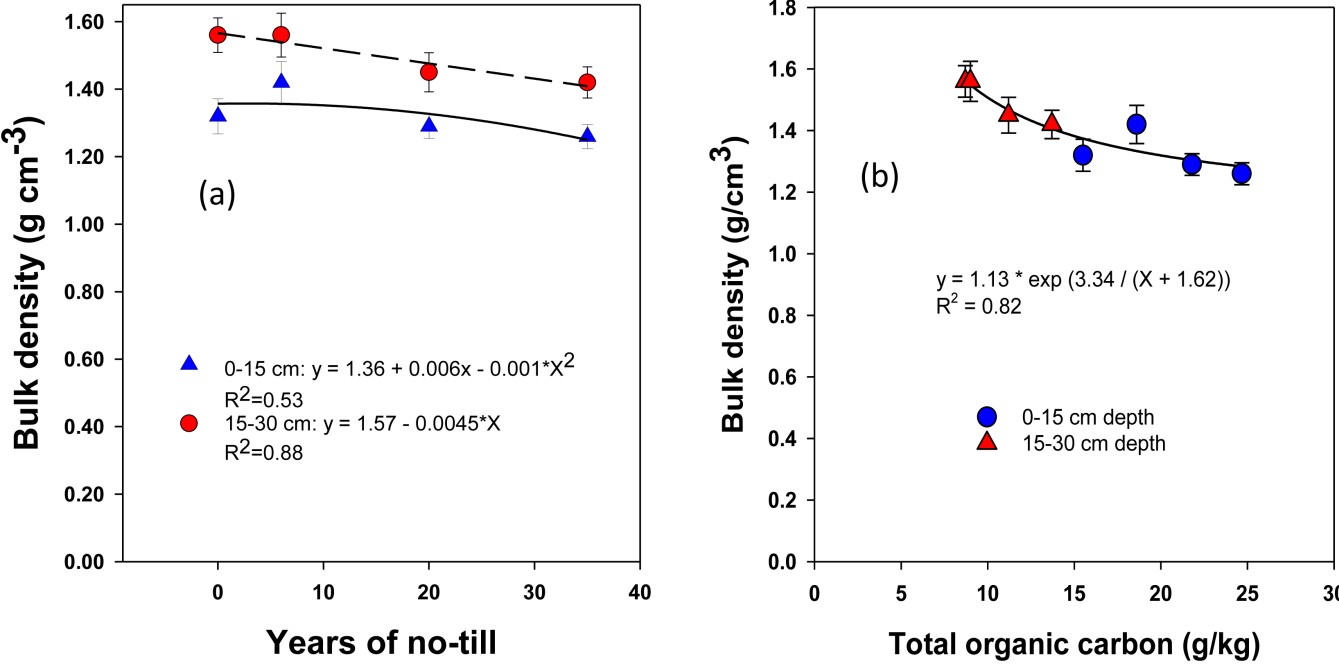

**Fig 3. Long-term continuous no-till (0, 6, 20, and 35 yr. ) effects on (a) bulk density and (b) the relationship between bulk density and total organic carbon at different soil depths under a rainfed corn-soybean system with cereal rye as a winter cover crop.** Treatment mean values are presented along with the standard error of the mean.

Based on fixed depth calculations, SOC was sequestered at significantly different rates across soil depths, with non-linear sequestration observed at a rate of 587.4±134.1 kg ha⁻¹ yr⁻¹ in the surface soil (0–15 cm depth) and a linear sequestration rate of 192±64 kg ha⁻¹ yr⁻¹ at a depth of 15–30 cm (Fig 4a). When applying the equivalent mass approach, which accounts for variations in bulk density and soil profile heterogeneity, SOC sequestration rates were slightly lower, with 467.5±168.3 kg ha⁻¹ yr⁻¹ at the 0–15 cm depth and 168.8±56.3 kg ha⁻¹ yr⁻¹ at 15–30 cm (Fig 4b). Similarly, total N accumulated non-linearly at a rate of 72.2±18.4 kg ha⁻¹ yr⁻¹ at the 0–15 cm depth and linearly at 15±5 kg ha⁻¹ yr⁻¹ at 15–30 cm, based on fixed depth calculations (Fig 4c). Using the equivalent mass approach, total N accumulated at rates of 52.4±19.9 kg ha⁻¹ yr⁻¹ at 0–15 cm and 17.3±5.8 kg ha⁻¹ yr⁻¹ at 15–30 cm (Fig 4d).

Based on the fixed depth approach, SMB was sequestered at rates that varied by soil depth. While at the surface soil (0–15 cm), SMB sequestration occurred non-linearly at a rate of 10.1±4.1 kg ha⁻¹ yr⁻¹ (Fig 5a), in contrast, at a depth of 15–30 cm, SMB sequestration occurred linearly at a rate of 1.6±0.5 kg ha⁻¹ yr⁻¹. When using the equivalent mass method, SMB sequestration was observed at a non-linear rate of 11.9±5.1 kg ha⁻¹ yr⁻¹ in the 0–15 cm depth and at a linear rate of 1.9±0.6 kg ha⁻¹ yr⁻¹ at the 15–30 cm depth (Fig 5b).

In contrast, active C sequestration occurred linearly in both the surface and subsurface soils based on the fixed depth approach (Fig 6). At the 0–15 cm depth, active C was sequestered at a rate of 5.3±1.8 kg ha⁻¹ yr⁻¹, while at the 15–30 cm depth, it was sequestered at a rate of 4.1±1.4 kg ha⁻¹ yr⁻¹ (Fig 6a). Using the equivalent mass approach, however, active C sequestration exhibited a non-linear pattern at a rate of 12.8±6 kg ha⁻¹ yr⁻¹ in the surface soil (0–15 cm), which was significantly higher than the fixed depth approach. At the 15–30 cm depth, active C sequestration occurred linearly at a rate of 6±2 kg ha⁻¹ yr⁻¹ (Fig 6b), showing a more moderate accumulation compared to the surface layer.

Similarly, passive C sequestration, which represents more stable forms of carbon, occurred non-linearly in the surface soil at a rate of 566.8±127.2 kg ha⁻¹ yr⁻¹ based on the fixed depth approach (Fig 6c). At deeper soil layers (15–30 cm),

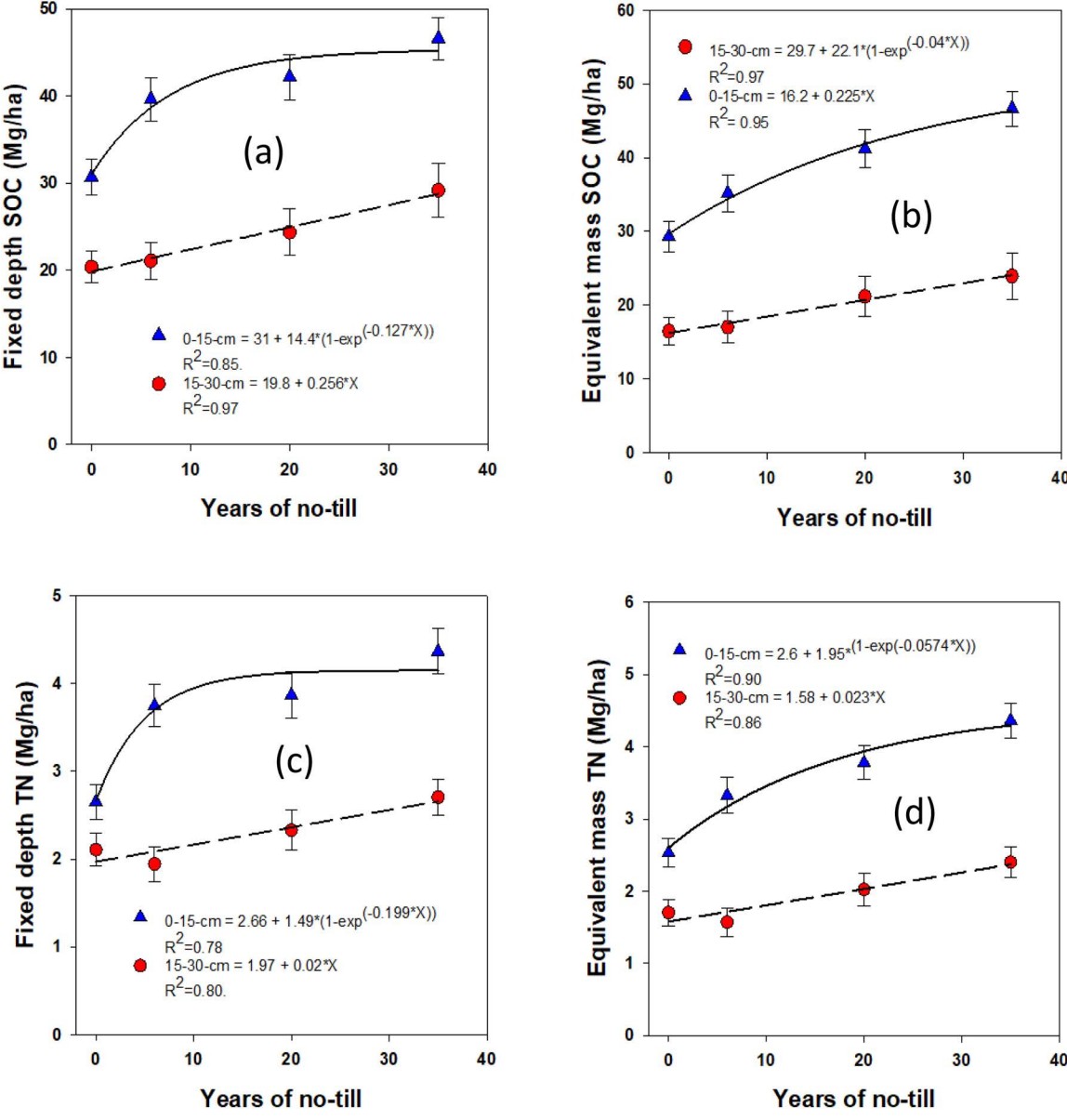

**Fig 4. Long-term continuous no-till effects (0, 6, 20, and 35 years) on total organic carbon (SOC) and total nitrogen (total N) stocks at different depths under a rainfed corn-soybean system with cereal rye as winter cover crop: Fixed-depth and equivalent mass approaches.** Treatment mean values are presented with the standard error of the mean.

passive C sequestration occurred linearly at a rate of 187.5±62.5 kg ha⁻¹ yr⁻¹. When using the equivalent mass approach, passive C sequestration rates were 457.2±164.1 kg ha⁻¹ yr⁻¹ at the 0–15 cm depth and 161.3±53.8 kg ha⁻¹ yr⁻¹ at 15–30 cm (Fig 6d).

The rates of POC sequestration exhibited distinct non-linear responses across soil depths, with marked differences observed between the 0–15 cm and 15–30 cm depths (Fig 7). Specifically, at the 0–15 cm depth, POC sequestration occurred at rates of 382.2±155.4 kg ha⁻¹ yr⁻¹ using the fixed depth approach, while the equivalent mass approach resulted in a sequestration rate of 352.5±132.1 kg ha⁻¹ yr⁻¹ (Fig 7a). At the 15–30 cm depth, however, both the fixed

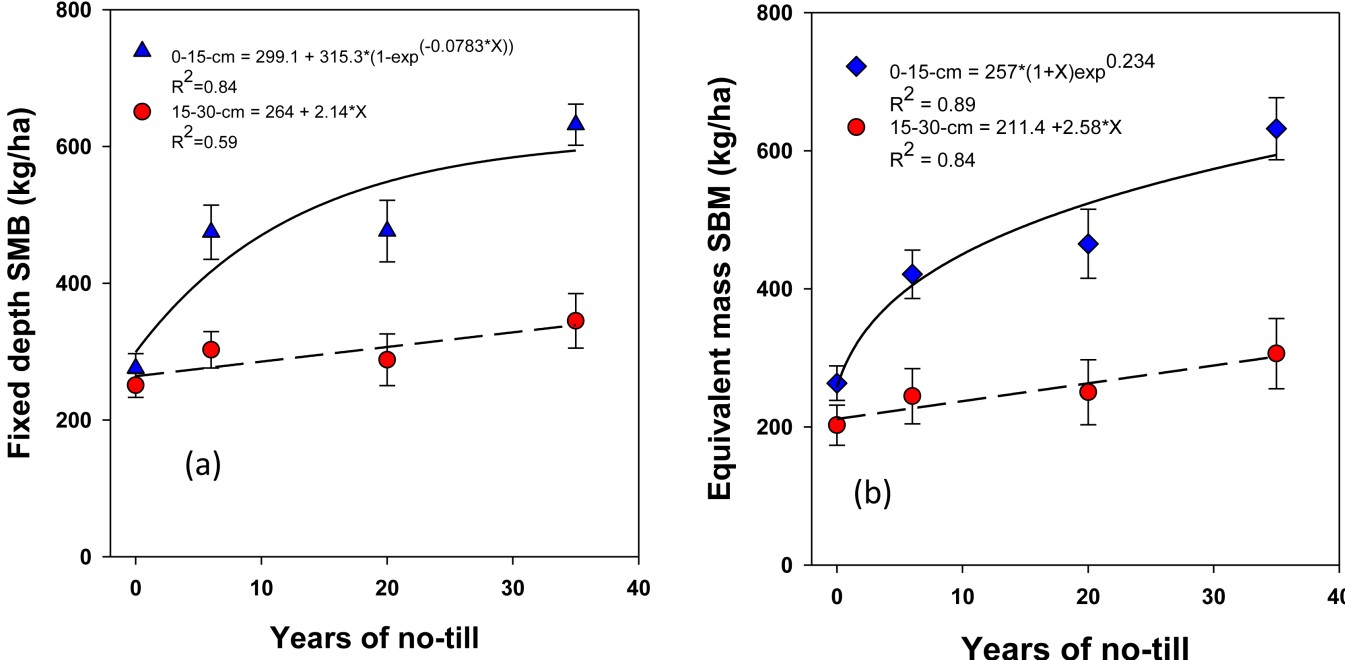

**Fig 5. Long-term continuous no-till (0, 6, 20, and 35 yr. ) effects on total microbial biomass carbon (SMB) stocks at different depths under a corn-soybean system with cereal rye as a winter cover crop: Fixed-depth and equivalent mass approaches.** Treatment mean values are presented with the standard error of the mean.

depth and equivalent mass approaches showed linear responses, with sequestration rates of $54 \pm 18$ kg ha⁻¹ yr⁻¹ and $54.8 \pm 18.3$ kg ha⁻¹ yr⁻¹, respectively (Fig 7b).

Similarly, PON accumulation followed a similar trend in terms of sequestration dynamics. Using the fixed depth approach, PON accumulated non-linearly at a rate of $14.1 \pm 5.5$ kg ha⁻¹ yr⁻¹ in the surface soil (Fig 7c). At the 15–30 cm depth, PON accumulation was observed to occur linearly at a rate of $4.3 \pm 1.6$ kg ha⁻¹ yr⁻¹. When the equivalent mass approach was applied, the rates of PON accumulation at the 0–15 cm depth were found to be $15.6 \pm 5.7$ kg ha⁻¹ yr⁻¹, slightly higher than the fixed depth method. (Fig 7d). At the 15–30 cm depth, PON accumulation rates were lower with the equivalent mass approach ($2.5 \pm 0.8$ kg ha⁻¹ yr⁻¹) than that of the fixed depth.

The storage capacity of SOM is finite and is significantly influenced by the soil's silt and clay content, as well as their intricate chemical and physical interactions with metal (oxyhydr)oxides [64–65]. These interactions play a critical role in determining the soil's long-term ability to retain and stabilize SOM. The dynamics of SOC) sequestration and total N accumulation in SOM exhibit notable depth-dependent responses. Surface soils (0–15 cm) are characterized by non-linear sequestration patterns, while deeper soils (15–30 cm) exhibit more linear accumulation trends.

The non-linear SOC sequestration and total N accumulation observed in surface soils (0–15 cm depth) reflect the rapid initial accumulation of SOM during the early years of NT adaptation. This phase is marked by enhanced inputs from unfragmented crop residues and roots, coupled with reduced disturbance, which collectively foster rapid increases in SOC and total N storage. Over time, however, the rates of accumulation slow as the soil approaches its maximum storage capacity. This is influenced by the saturation of stabilization sites on silt, clay, and metal (oxyhydr)oxides, as well as shifts in microbial activity and decomposition dynamics.

In contrast, the linear increases in SOC and total N observed in deeper soils (15–30 cm depth) suggest a more gradual and continuous incorporation of C and N in SOM. This accumulation is driven by slower biological

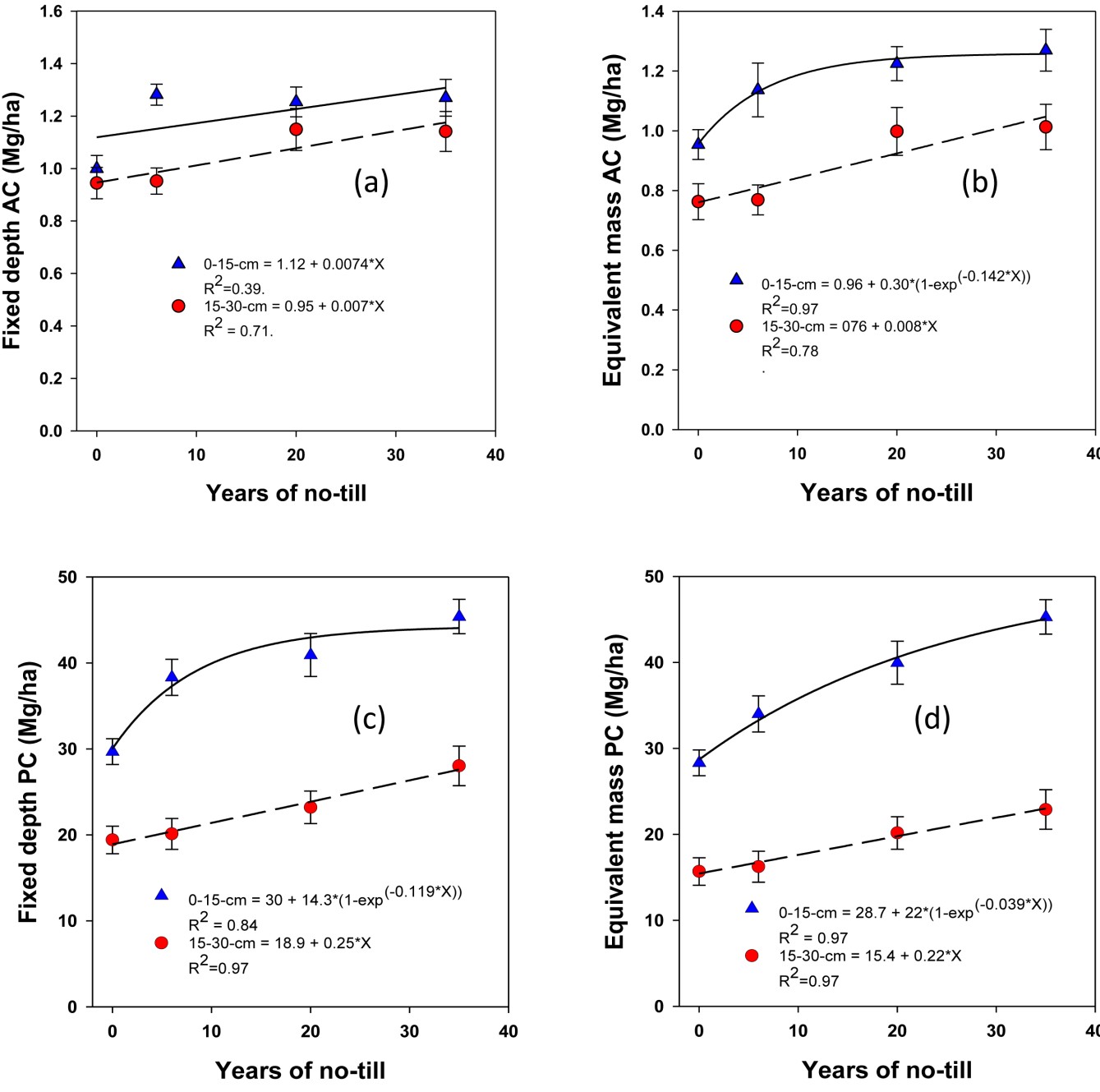

**Fig 6. Long-term continuous no-till effects (0, 6, 20, and 35 years) on active carbon (active C) and passive carbon (passive C) stocks at different depths under a rainfed corn-soybean system with cereal rye as winter cover crop: Fixed-depth and equivalent mass approaches.** Treatment mean values are presented with the standard error of the mean.

processes, such as the downward movement of dissolved organic matter and fine particulate organic material, as well as root turnover and microbial activity in subsurface layers. Unlike the surface soils, where SOC and total N pools experience rapid initial changes, deeper soils provide a more consistent and sustained capacity for sequestration, albeit at lower rates.

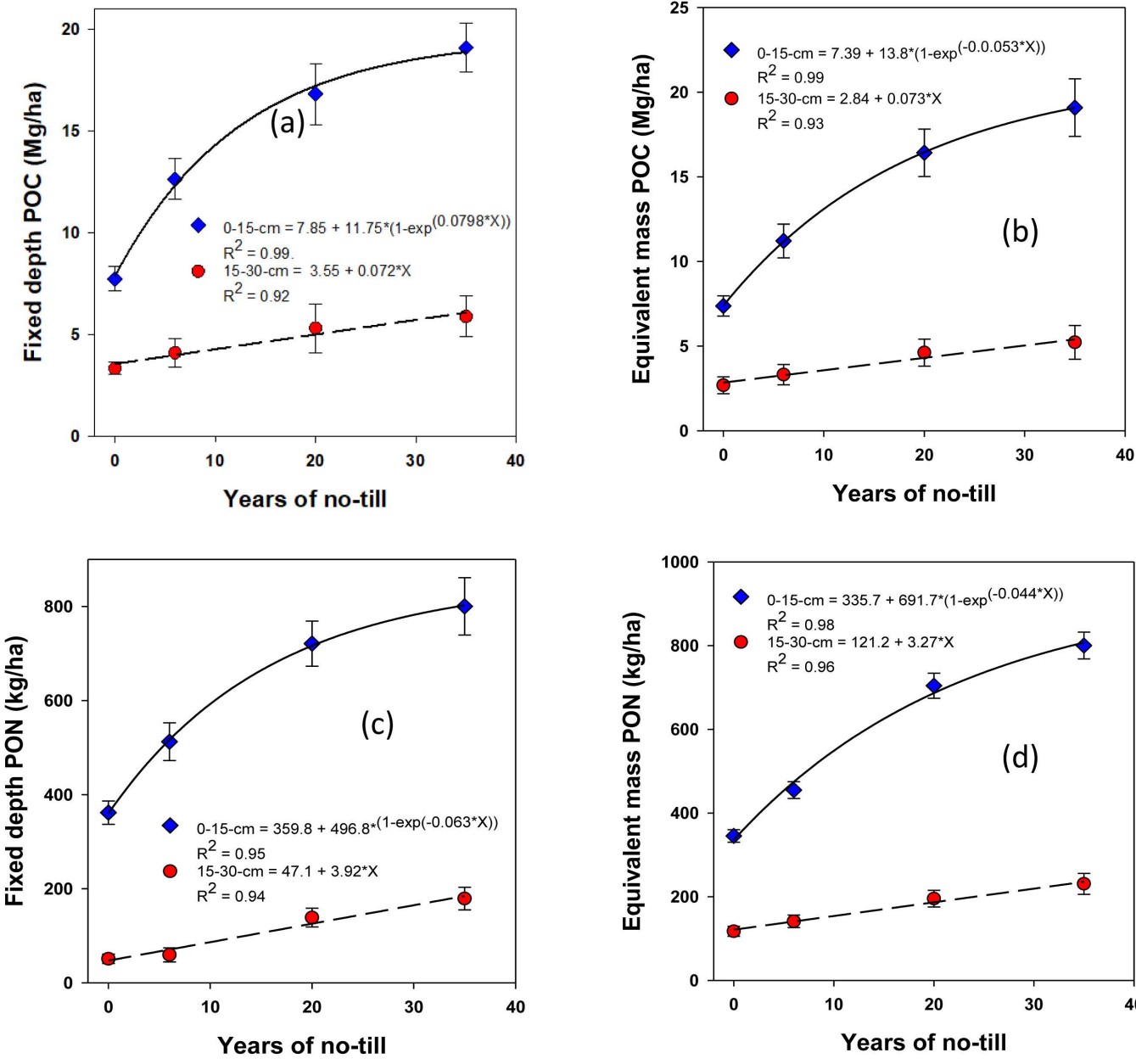

**Fig 7. Long-term continuous no-till (0, 6, 20, and 35 yr.) effects on particulate organic carbon (POC) and particulate organic nitrogen (PON) stocks at different depths under a rainfed corn-soybean system with cereal rye as winter cover crop: Fixed-depth and equivalent mass approaches.** Treatment mean values are presented with the standard error of the mean.

Over time, SOC sequestration and total N accumulation in both surface and subsurface pools begin to plateau as the soil approaches its storage capacity. This plateau is particularly evident in surface soils under long-term NT management, where the saturation of stabilization mechanisms and physical protection of SOM limit further increases. Understanding these depth-dependent patterns is critical for optimizing NT and other conservation practices to maximize SOC and total N sequestration while maintaining soil health and resilience.

Our results showed that at 0–15 cm depth, SOC sequestration and total N accumulation in different pools plateaued after about 15 yr. of NT, whereas at 15–30 cm depth, they continued to increase linearly, suggesting further accumulation even beyond 35 yr. While other studies report minor or insignificant NT impact on SOC sequestration [3,31,65,66], our results align with those indicating SOC sequestration and stoichiometric N accumulation under diverse conditions [16,32,41,67]. Switching from plowing to NT increased SOC stock by 570 kg ha$^{-1}$ yr$^{-1}$; while others reported that SOC sequestration linearly peak between > 0 and 10 yr. and decline after 15–20 yr. [29,32,68]. It is reported that sequestration rates of biological pool of SOC (SMB) were 22, 13, 7, and 3 kg ha$^{-1}$ yr$^{-1}$ at the 0–7.5, 7.5–15, 15–22.5, and 22.5-30-cm depths, respectively for the first 10 yr. of NT [29]. During the same time, the macro-aggregate protected C sequestration rates were 92, 63, 47, and 37 kg ha$^{-1}$ yr$^{-1}$, respectively, at the same depths. By 20 yr. of NT, SMB and macroaggregate protected C sequestration rates plateaued at the surface. Similarly, another study reported that SOC sequestration under NT decreased over time with initially higher rates at 740 kg ha$^{-1}$ yr$^{-1}$ in the 3–10 yr. followed by 260 kg ha$^{-1}$ yr$^{-1}$ in the 11–20 yr, and 95 kg ha$^{-1}$ yr$^{-1}$ for longer than 20 yr in the medium textured prairie soils in Canada [69].

## Relationship between fixed depth and equivalent mass soil carbon and nitrogen stocks

The stocks of SOC and total N pools showed significant linear relationships using both the fixed depth and equivalent mass approaches, with the fixed depth method consistently overestimating values (Figs 8–10). The relationship for SOC stock was highly significant ($R^2$ = 0.98) with a slope of 1.12 ± 0.06 at $p < 0.0001$. However, the fixed depth approach overestimated SOC by 6.61 ± 2.01 Mg ha$^{-1}$ at $p = 0.017$ (Fig 8a). Similarly, total N stock exhibited a comparable relationship ($R^2$ = 0.98, slope 1.11 ± 0.06 at $p < 0.0001$, with fixed depth overestimating by 576.5 ± 175 kg ha$^{-1}$ at $p = 0.016$ (Fig 8b).

The fixed depth active C stock accounted for 88% of the variation ($R^2$) in equivalent mass active C stock, with a slope of 1.26 ± 0.18 at $p = 0.004$, overestimating by 40.4 ± 10.1 kg ha$^{-1}$ at $p = 0.09$ (Fig 8c). Likewise, passive C stock was significantly overestimated by 6.2 ± 1.9 Mg ha$^{-1}$ at $p = 0.017$, with $R^2$ = 0.98 and a slope of 1.11 ± 0.06 at $p < 0.0001$ (Fig 8d).

The relationship between SMB stock as calculated by the fixed depth and equivalent mass approaches was highly significant ($R^2$ = 0.98) with a slope of 1.09 ± 0.056 ($p < 0.0001$). However, the fixed depth approach consistently overestimated SMB by 65.8 ± 22.5 kg ha$^{-1}$ ($p = 0.027$) compared to the equivalent mass method, as shown in Fig 9.

Similarly, overestimations were observed for POC and PON stocks when using the fixed depth approach. For POC, the fixed depth method overestimated stock by 842 ± 142 kg ha$^{-1}$ ($p = 0.025$), with a highly significant linear relationship ($R^2$ = 0.99, slope 1.02 ± 0.03, $p < 0.0001$) as shown in Fig 10a. For PON, the fixed depth approach accounted for 99% of the variation ($R^2$) in equivalent mass PON stock, with a slope of 1.03 ± 0.024 ($p < 0.0001$). Despite this high correlation, the fixed depth method overestimated PON by 37.8 ± 11.3 kg ha$^{-1}$ ($p = 0.015$), as shown in Fig 10b.

Results showed that the fixed depth approach failed to account for ρb differences, overestimated SOC and total N pools, particularly in surface soils where stratification is more pronounced under long-term NT, when compared to the equivalent mass method. Several studies have reported that SOC and N stocks calculated using the equivalent mass approach are more realistic, as the fixed depth method often leads to inaccuracies [40–42,70]. Specifically, fixed depth approaches overestimate SOC sequestration in soils with higher ρb (associated with compaction) and underestimate it in soils with lower ρb [71]. Our findings validate that adjusting for soil mass equivalence provides more accurate estimations of SOC and N stocks across different pools, avoiding the overestimation inherent in fixed depth methods.

## Conclusions

Long-term continuous NT practice in a rainfed corn-soybean rotation significantly increased SOC, total N, SMB, active C, passive C, POC, and PON contents. It also improved biological properties especially microbial C-use efficiency (reduced $qCO_2$) compared to CT. These positive effects were more evident in the surface soil (0–15 cm depth) than in the

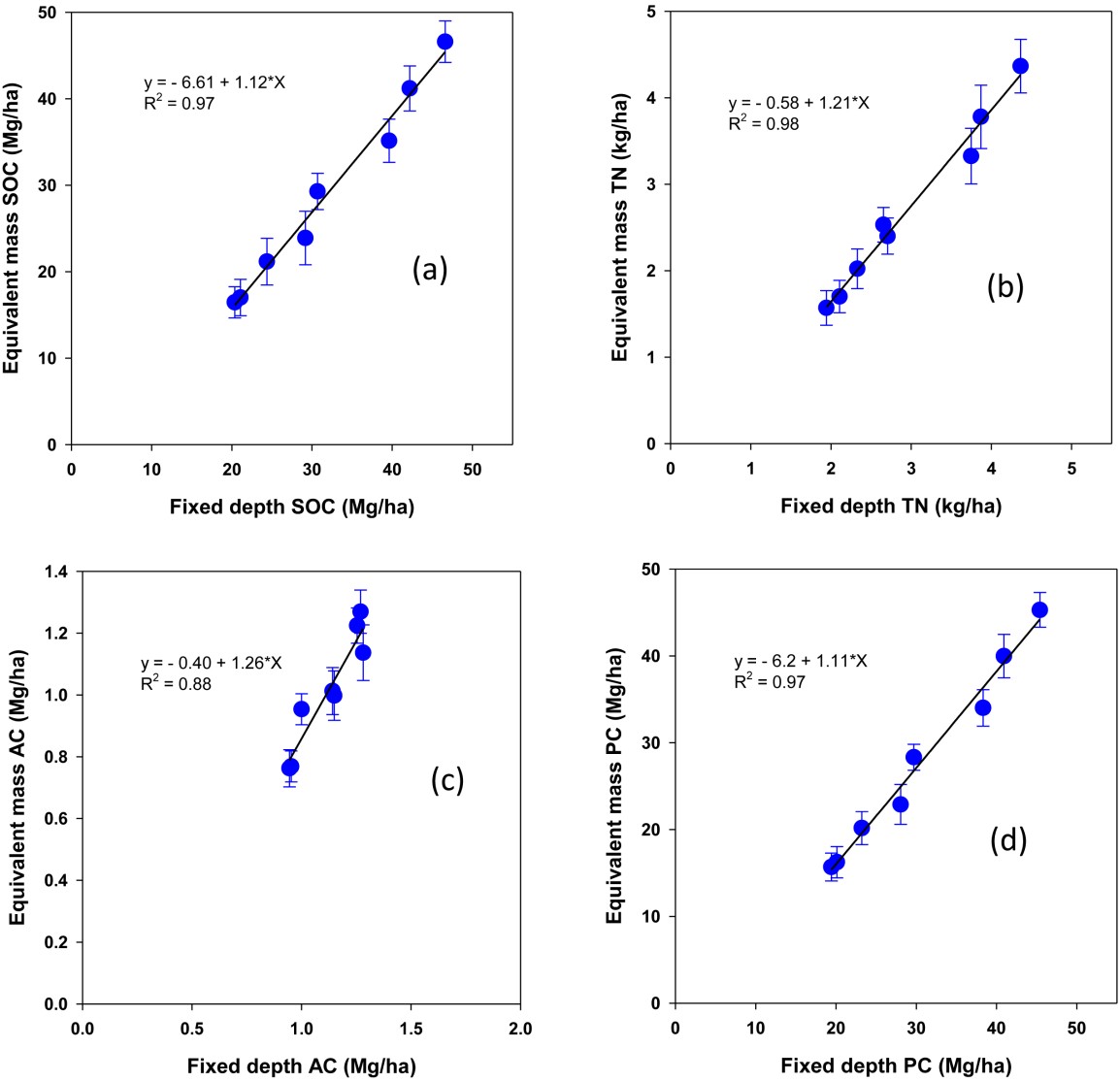

**Fig 8. Relationship between fixed-depth and equivalent mass stocks of total organic carbon (SOC), total nitrogen (total N), active carbon (AC), and passive carbon (passive C) under a rainfed corn-soybean system with cereal rye as winter cover crop.** Treatment mean values are presented with the standard error of the mean.

subsurface soil (15–30 cm depth). A higher proportion of passive C compared to active C suggests an increase in the stable form of SOC under long-term NT, which helps to sequester C in SOM. While short-term NT resulted in slightly higher compaction (ρb), the NT$_{35}$ significantly reduced soil compaction at both depths compared to CT. The SOC sequestration and total N accumulation in different pools under NT were nonlinear at the surface soil and showed a linear response at subsurface soil when calculated using both fixed depth and equivalent mass approaches. The stocks of SOC and total N had significant linear relationships using both fixed depth and equivalent mass approaches, with fixed depth consistently overestimating SOC sequestration and total N accumulation rates. Results of our on-farm long-term NT study has demonstrated that adjusting for soil mass equivalence provides a more realistic estimation and prediction of SOC sequestration and total N accumulation in different pools under a rainfed corn-soybean system.

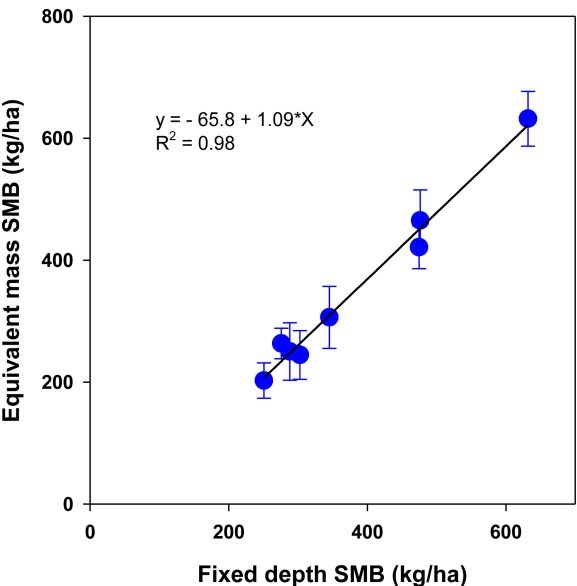

**Fig 9. Relationship between fixed-depth and equivalent mass stocks of total microbial biomass carbon (SMB) under a rainfed corn-soybean system with cereal rye as winter cover crop.** Treatment mean values are presented with the standard error of the mean.

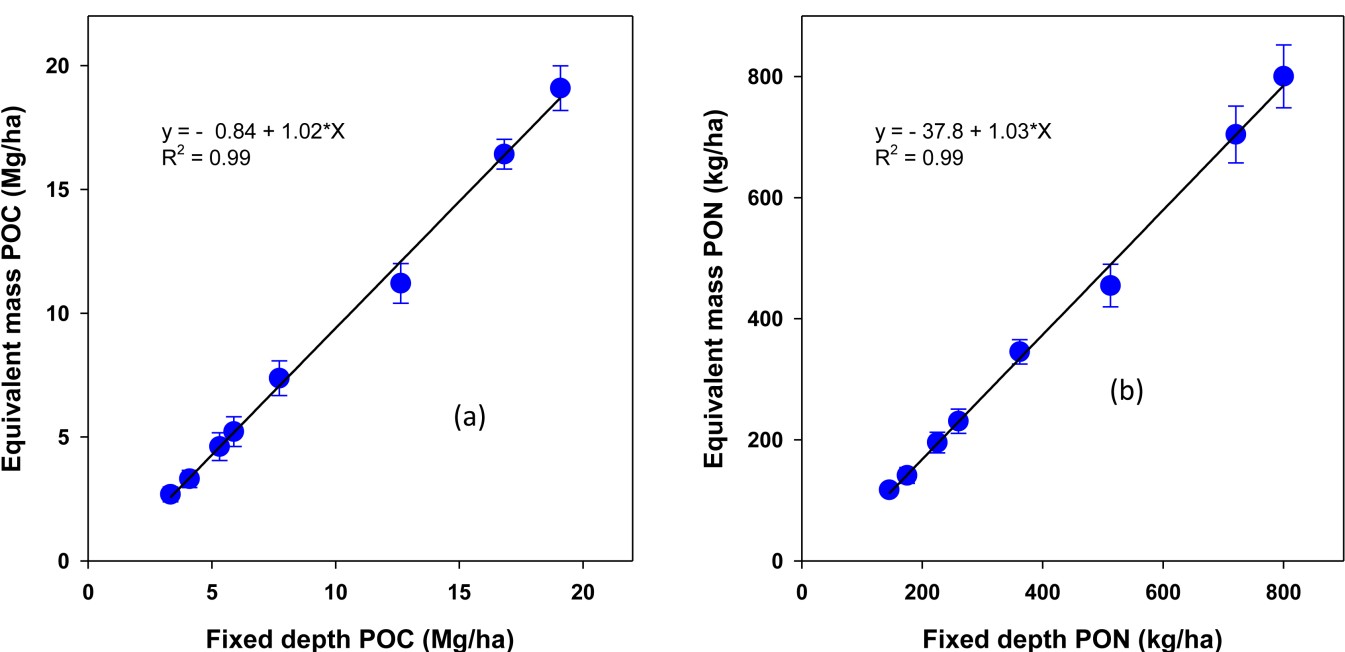

**Fig 10. Relationship between fixed-depth and equivalent mass stocks of particulate organic carbon (POC) and particulate organic nitrogen (PON) under a rainfed corn-soybean system with cereal rye as winter cover crop.** Treatment mean values are presented with the standard error of the mean.

## Supporting Information

**S1 File.** Long-term continuous no-till management (0, 6, 20, and 35 years) effects on microbial biomass, biological activity, organic carbon and nitrogen pools, and associated soil properties across various depths under a rainfed corn–soybean rotation with cereal rye (Secale cereale) as a winter cover crop at Brandt's farm in Carroll, Ohio, USA. (XLSX)

## Acknowledgments

The authors extend their deepest gratitude to the late David Brandt, the esteemed owner of Brandt Family Farm located in Carroll, Fairfield County, in central Ohio, United States. His generosity in allowing access to his farm and providing comprehensive logistical support for soil sampling was invaluable to this research. While David Brandt's passed away last year, his contributions and commitment to sustainable agriculture have left a lasting legacy, and his support has been crucial in advancing our understanding of no-till and soil health.

## Author contributions

**Data curation:** Maninder K. Khosa, Khandakar R. Islam.

**Formal analysis:** Maninder K. Khosa, Nataliia O. Didenko, Kenan Barik, Ekrem Aksakal.

**Investigation:** Maninder K. Khosa, Nataliia O. Didenko, Kenan Barik, Ekrem Aksakal.

**Methodology:** Khandakar R. Islam, Maninder K. Khosa, Nataliia O. Didenko.

**Project administration:** Khandakar R. Islam.

**Resources:** Khandakar R. Islam.

**Software:** Mohammad MR. Jahangir, Khandakar R. Islam.

**Supervision:** Khandakar R. Islam.

**Validation:** Maninder K. Khosa, Mohammad MR. Jahangir, Khandakar R. Islam.

**Visualization:** Mohammad MR. Jahangir, Khandakar R. Islam.

**Writing:** Original draft: Maninder K. Khosa, Ekrem Aksakal, Kenan Barik.

**Writing:** Review & editing: Khandakar R. Islam, Mohammad MR. Jahangir.

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
