## [Decision Letter · Decision Letter 0]

15 Sep 2024

PONE-D-24-34295Long-term, Continuous No-Till Corn-Soybean Systems: Examining Soil Carbon Sequestration and Nitrogen Accumulation Across Various PoolsPLOS ONE

Dear Dr. Islam,

Thank you for submitting your manuscript to PLOS ONE. After careful consideration, we feel that it has merit but does not fully meet PLOS ONE’s publication criteria as it currently stands. Therefore, we invite you to submit a revised version of the manuscript that addresses the points raised during the review process.

We look forward to receiving your revised manuscript.

Kind regards,

Sushanta Kumar Naik, PhD

Academic Editor

PLOS ONE

Journal Requirements: When submitting your revision, we need you to address these additional requirements. 1. Please ensure that your manuscript meets PLOS ONE's style requirements, including those for file naming. The PLOS ONE style templates can be found at https://journals.plos.org/plosone/s/file?id=wjVg/PLOSOne_formatting_sample_main_body.pdf and https://journals.plos.org/plosone/s/file?id=ba62/PLOSOne_formatting_sample_title_authors_affiliations.pdf 2. Thank you for stating the following in your Competing Interests section: "Not applicable" Please complete your Competing Interests on the online submission form to state any Competing Interests. If you have no competing interests, please state ""The authors have declared that no competing interests exist."", as detailed online in our guide for authors at http://journals.plos.org/plosone/s/submit-now   This information should be included in your cover letter; we will change the online submission form on your behalf. 3. In the online submission form, you indicated that "Upon request, data will be available." All PLOS journals now require all data underlying the findings described in their manuscript to be freely available to other researchers, either 1. In a public repository, 2. Within the manuscript itself, or 3. Uploaded as supplementary information.This policy applies to all data except where public deposition would breach compliance with the protocol approved by your research ethics board. If your data cannot be made publicly available for ethical or legal reasons (e.g., public availability would compromise patient privacy), please explain your reasons on resubmission and your exemption request will be escalated for approval. 4. When completing the data availability statement of the submission form, you indicated that you will make your data available on acceptance. We strongly recommend all authors decide on a data sharing plan before acceptance, as the process can be lengthy and hold up publication timelines. Please note that, though access restrictions are acceptable now, your entire data will need to be made freely accessible if your manuscript is accepted for publication. This policy applies to all data except where public deposition would breach compliance with the protocol approved by your research ethics board. If you are unable to adhere to our open data policy, please kindly revise your statement to explain your reasoning and we will seek the editor's input on an exemption. Please be assured that, once you have provided your new statement, the assessment of your exemption will not hold up the peer review process. 

**Additional Editor Comments:**

Write the introduction in a cone approach. Highlight the present problem throughout the globe-regional level. How the present approach will overcome the problem with details.

Mention the hypothesis of your study.

Abstract to be re-written as reviewers pointed out.

Check all the captions of figures and tables.

The references for various methods adopted are missing in the methodology section.

The statistical analysis followed in all the tables are not giving a clear picture and should be modified. Depth effects are missing. The interaction table is not properly analyzed.

Results and discussion to be re-written as pointed out be reviewer.

Reviewers' comments:

Reviewer's Responses to Questions

**Comments to the Author**

1. Is the manuscript technically sound, and do the data support the conclusions?

Reviewer #1: Partly

Reviewer #2: Yes

2. Has the statistical analysis been performed appropriately and rigorously? 

Reviewer #1: No

Reviewer #2: Yes

3. Have the authors made all data underlying the findings in their manuscript fully available?

Reviewer #1: Yes

Reviewer #2: Yes

4. Is the manuscript presented in an intelligible fashion and written in standard English?

Reviewer #1: No

Reviewer #2: Yes

5. Review Comments to the Author

Reviewer #1: Comments to the authors of “Long-term, Continuous No-Till Corn-Soybean Systems: Examining Soil Carbon Sequestration and Nitrogen Accumulation Across Various Pools”. While the topic is interesting, the authors need to take a major revision of the manuscript to significantly increase the readability of the study. Also, the concept of high SOC and total N with medium to long term no-tillage has been comprehensively investigated, the authors should focus on adding the novelty to this study, specifically to the introduction section. Pls consider adding a section on soil N pools as well to the introduction. The current introduction only focuses on soil C. The data analysis section should be revised, and soil depth needs to be accounted for as a repeated variable. Essentially, the statistical model should be a repeated measures repeated by soil depth. Also, the soil depth effects are missing from the tables. The authors have only presented tillage effects and interaction of tillage with depth. But the soil depth differences need to be presented (not the pooled data). The results and discussion section need to be significantly revised and reorganized. Its helpful to merge the discussion of the results along with the presentation of results. Right now, the results are all written at once and then discussed later, which is not helpful. Pls see the attached pdf file for the specific comments.

Reviewer #2: In no till planting, there are less steps and cost involved, thus productivity is higher. The soil does not have to be tilled. Instead, the seeds are planted through the remains of previous crops by planters or drills. in present study, authors concluded from their study that long-term continuous NT practice in a rainfed corn-soybean rotation with cereal rye as a winter cover crop significantly increased soil bacteria, active carbon, POC, passive C, SOC, PON, and total N. It also improved biological efficiency compared to CT. These positive effects were more evident in the surface soil than in the subsurface soil. A higher proportion of passive C compared to active C in SOC suggests an increase in the stable form of SOC under NT. While transitional NT resulted in slightly higher compaction, the soil became more porous under long-term NT at both depths compared to CT. SOC sequestration nd stoichiometric N accumulation in different pools under NT were nonlinear at the surface and showed a linear response at subsurface depth when calculated using both fixed depth and equivalent mass approaches. Overall the paper is written well, however, following need to be addressed.

-there are number of mistakes in english for grammar and spelling.

-Quality of figures is not good.

6. PLOS authors have the option to publish the peer review history of their article (what does this mean? ). If published, this will include your full peer review and any attached files.

**Do you want your identity to be public for this peer review?** For information about this choice, including consent withdrawal, please see our Privacy Policy .

Reviewer #1: **Yes: ** Inderjot Chahal

Reviewer #2: No

---

## [Author Response · Author response to Decision Letter 1]

26 Jan 2025

A revised response to reviewers' comments was added with a statement "The authors have declared that no competing interests exist", as an attached file.

---

## [Editor Report · Decision Letter 1]

30 Mar 2025

Long-term, Continuous No-Till Corn-Soybean Systems: Examining Soil Carbon Sequestration and Nitrogen Accumulation Across Various Pools

PONE-D-24-34295R1

Dear Dr. Islam,

We’re pleased to inform you that your manuscript has been judged scientifically suitable for publication and will be formally accepted for publication once it meets all outstanding technical requirements.

Kind regards,

Sushanta Kumar Naik, PhD

Academic Editor

PLOS ONE
---

## [Editor Report · Acceptance letter]

PONE-D-24-34295R1

PLOS ONE

Dear Dr. Islam,

I'm pleased to inform you that your manuscript has been deemed suitable for publication in PLOS ONE. Congratulations! Your manuscript is now being handed over to our production team.

Kind regards,

on behalf of

Dr. Sushanta Kumar Naik

Academic Editor

PLOS ONE